# Is Telemedicine in Primary Care a Good Option for Polish Patients with Visual Impairments Outside of a Pandemic?

**DOI:** 10.3390/ijerph19116357

**Published:** 2022-05-24

**Authors:** Katarzyna Weronika Binder-Olibrowska, Magdalena Agnieszka Wrzesińska, Maciek Godycki-Ćwirko

**Affiliations:** 1Department of Psychosocial Rehabilitation, Faculty of Health Sciences, Medical University of Lodz, Lindleya 6, 90-131 Lodz, Poland; magdalena.wrzesinska@umed.lodz.pl; 2Centre for Family and Community Medicine, Faculty of Medical Sciences, Medical University of Lodz, Kopcińskiego 20, 90-153 Lodz, Poland; maciej.godycki-cwirko@umed.lodz.pl

**Keywords:** blindness, digital literacy, general practice, health equity, healthcare needs, primary healthcare, teleconsultations, telemedicine, visual impairment

## Abstract

With the proliferation of telemedicine during the COVID-19 pandemic, attention began to turn to the risk of health disparities associated with its use among people with disabilities. Therefore, the present study investigates the level of interest in using teleconsultations (TCs) in primary healthcare among people with visual impairments (PVIs) and identifies their motivations and needs. A total of 219 Polish PVIs were surveyed using a combination of closed and open questions. About 50% of the respondents expressed interest in using TCs. The factor most closely related to the willingness to use TCs was age. The predominant reason for using TCs was to obtain a prescription or referral, and the most highlighted need was the possibility to choose between a TC and an in-person visit. The blind and poor-sighted participants differed in some regards. Our study indicates that TCs, under some conditions, may be a beneficial option for PVIs, and provides some directions for its effective implementation.

## 1. Introduction

While the series of restrictions and the need for self-isolation associated with the COVID-19 pandemic has distanced people from each other, this distancing has also demonstrated that a lack of direct contact is not always an obstacle in providing healthcare. One particular example of this is the use of telemedicine (TM).

### 1.1. Telemedicine—Role, Advantages and Disadvantages

The objectives of TM as defined by the World Health Organization (WHO) include the diagnosis, treatment, and prevention of disease and injury by all health professionals using information and communication technology (ICT), where distance is a critical factor, and its primary goal is to improve human health [1]. TM plays a crucial role in the sustainability of the public health system as it is applicable and clinically effective in many areas of medicine, such as ophthalmology, cardiology, dermatology, neurology, multidisciplinary care [2], and primary care [3]. It enables the continuation of patient care and increases the level of safety for both patients and staff by reducing the risk of spreading infections during pandemics [4,5,6]; moreover, TM has been found to decrease mortality and hospitalizations due to chronic diseases (NCDs), such as diabetics and chronic heart failure, to enhance mental health and a healthy lifestyle and may bring cost and time savings to patients and providers [7,8]. On the other hand, there are some limitations to the effective usage of TM [9], which can be classified as: legal, financial, ICT-technology, and awareness-related barriers [10]. In addition, certain groups, including people with disabilities (PwDs), are more likely to face specific barriers in the use of TM, such as insufficient internet infrastructure, the “digital divide”, or the lack of assistive devices. Therefore, with the growing popularity of e-solutions, an urgent need has emerged to better comprehend how TM implementation can best accommodate the needs of people living with disabilities [11,12,13,14].

### 1.2. Visual Impairment and Health

It has been proposed that individuals with sensory disabilities, including people with visual impairments (PVIs), may experience particular difficulties when using TCs [15,16]. The PVIs group represents a significant part of the world population: in 2020, 43.3 million people were registered blind, 295 million had moderate to severe visual impairment (VI), and 258 million had a mild VI. It is predicted that by 2050, 61 million people will be blind, 834 million will have mild to severe VI [17], and half of the world’s general population will be affected by myopia (the leading cause of VI) [18]. VI may contribute to adverse health outcomes by affecting biopsychosocial functioning [19]. Compared to the general population, PVIs are at higher risk of multimorbidity [20,21,22], mental health problems [23,24,25], falls, and resulting injuries [26]. Moreover, during the COVID-19 pandemic, it was also noted that PVIs perceive themselves as more vulnerable to infection [27]. Hence, they have higher health risks and needs, and such difficulties may further aggravate their health anxiety [27]. Additionally, PwDs have limited access to medical care due to higher external costs, poorer access to transportation, and less access to rehabilitation and technology, and these inequalities in healthcare are exacerbated by the lack of adequate training or experience among medical personnel in responding to the needs of people with severe vision loss [28]. Although TM seems to have some potential as a solution here and it is now a frequent subject of scientific research, relatively few studies have examined its relationship with people with disabilities (PwDs); nevertheless, some have pointed out that PwDs are vulnerable to health inequalities, and there is a need to better understand how the implementation of TM can be best suited to their needs [29].

### 1.3. Polish Primary Care and TM

Our study focuses on primary healthcare (PHC), as it is the patient’s first point of contact with the health services [30,31,32,33,34] and it is able to meet the health needs of people in about 80–90% cases [35]. Polish PHC services are financed from public funds; they include prevention, diagnostics, treatment, nursing, and rehabilitation, and the PHC center typically comprises a team consisting of a primary care physician (PCP), a nurse, and a midwife, who are chosen by the recipient. PHC services are provided to eligible patients at their place of residence, in outpatient clinics, or, in justified cases, at the patient’s home [36].

In Poland, TM was authorized in 2015 [37], and PHC clinics have been obliged to provide it to patients since the beginning of 2020 [38]. It gained significant popularity during the COVID-19 pandemic. An August 2020 report found that 80% of visits to Polish PHCs since the beginning of the COVID-19 pandemic were TCs [39]. A similar trend in TC uptake has also been observed in other countries, including Australia, Canada, and the United States [40,41,42]. In August 2020, a study of Polish patients found half to be satisfied with TCs [39]; however, later studies revealed more negative attitudes, particularly among those with chronic diseases or experiencing emergencies, who complained that TCs were replacing visits instead of supplementing them [43,44]. However, the preferences and experiences of Polish PwDs in this context remain unstudied. Only a single review has so far examined the opportunities, threats, and challenges connected with TC utilization in a population with VI, and this was prepared as a background for the current study [45].

### 1.4. Objectives

Taking into account two of the key strategic goals of the Polish health policy defined in 2019, which are the development of TM and the strengthening of care for PwDs [46], our study addresses the following research questions:Are PVIs interested in PHC TCs?Is their interest in remote PHC consultations associated with socio-demographic variables, the level of vision deficits, independence in mobility, having NCDs or additional disabilities, previous experience in TM, and digital access and skills?What are the reasons for PVIs to use TCs?Do persons with blindness and low vision differ in their motives for using TCs?What are the needs of PVIs when utilizing TCs?Are there any differences in the needs of the poor sighted and the blind?


## 2. Materials and Methods

### 2.1. Study Design and Participants

A cross-sectional exploratory study design was used to gather opinions from a significant PVIs’ population and adjust to the diversity of their functioning and difficulties in direct contact arising from pandemic restrictions. A survey was carried out from May to July 2021 among poor-sighted and blind people throughout Poland. The inclusion criteria were as follows: voluntary written or verbal consent to participate in the study, self-declaration of being at least 18 years old, and having VI. Failure to meet any of the inclusion conditions or difficulty understanding items—reported by the respondent or noticed by the interviewer—were exclusion criteria.

No register of people belonging to the target population exists in Poland, neither in national institutions nor in clinics. Data collected by non-public organizations were unavailable due to data protection. Therefore, participants were recruited using accidental selection and snowball sampling: the former refers to those respondents who volunteered for the study after encountering an invitation on an internet forum, school, association, sports club, or clinics for PVIs. Information about the study was posted on flyers and websites and distributed by representatives of the institutions listed above. Following this, a snowball method allowed for the inclusion of people who learned about the study from other respondents.

Several forms of recruitment and data collection were planned to engage as many people as possible with different levels of VI, digital skills, and place of residence during the planned time of the study. The recruitment protocol also complied with the SARS-CoV-2 pandemic restrictions, minimizing the need for direct contact. Participants responded via Google Form, e-mail, paper, or phone. The paper-pencil questionnaires and print versions of the announcement were prepared in enlarged font and included a QR code to read the information electronically. None of the respondents took the opportunity to use the Braille questionnaire. Participants could register for the telephone survey through a Google Form or the phone number provided in the survey announcement. The telephone interviewer was trained to communicate appropriately with PVIs and not to influence their answers. The respondents were assured of anonymity.

Most participants learned about the study through PVIs associations (37% of study group). Among the other groups, those gathered by internet forums, previous participants, and “*in another way*” constituted about 20% each of the total. Only two people completed the survey at an eye clinic or department, and eight participants were encouraged by a disability representative at a university.

### 2.2. Measures

The survey used a self-administered questionnaire complied for the purpose of this study. It comprised a set of three structured questions and two open-ended questions on the subject of TM in PHC (which can be found in the Questionnaire Sheet in Appendix A) and questions regarding the respondents’ characteristics.

Firstly, the participants were asked to indicate in which situations TCs might be a better option for poor-sighted and blind people than an in-person visit to the PHC. Based on a previous literature review [45], a list of eight responses to this question was created (presented in Figure 1). Every participant could indicate more than one answer. To obtain more details and spontaneous opinions from the study population, the following additional open question was used: *In which other situations would you like to use TCs at PHC?*

Then, participants indicated their needs in the context of TCs using a list of seven possible answers created on the basis of a previous literature review [45] (Figure 2). To identify issues not covered by the closed questions in this section, a second open-ended question was added: *What else do you think is vital for the blind and partially sighted persons in the context of teleconsultation with primary care physician?* This question allowed the participants to develop a beyond-personal perspective so it could provide knowledge about the needs of the target population and not only one specific respondent.

Finally, the respondents indicated whether they were interested in using TCs, with the response given on the following 5-point Likert scale: *definitely yes/rather yes/hard to say/rather not/definitely not*. For the purposes of statistical analysis, the responses were aggregated to *yes, hard to say, no*. This question was provided at the end of the questionnaire connected with TM, as the response may be influenced by the foregoing questions.

The survey also collected sociodemographic data (sex, age, level of education, place of residence, income, other persons with disabilities or care needs in the respondent’s household). Furthermore, participants also reported any NCD (yes/no question) or additional disabilities (multiple answers could be selected from *no other disability, hearing impairment, mobility impairment,* and *another disability;* the responses were reduced to *yes* and *no*).

The respondents indicated whether they had held a TC at the PHC in the six months before the study, and whether they had any experiences in using a cell phone, computer, and the internet (on the scale: *yes/no*). As only one person reported not using a mobile phone, this variable was not included in the analysis. The participants also assessed their digital skills, such as their computer and internet proficiency on the following scale: *lack of ability, low, average,* and *high level*.

Additionally, the respondents also indicated how they travel in public spaces based on the following list, with more than one answer possible: *on my own, without any outside help or assistance, with the help of a guide/assistant, a white cane, a guide dog, the use of GPS,* or *other*. The only response considered as independent mobility for the purposes of the further analysis was *independently, without external assistance or aids*. These indicators were then related to the level of interest in TCs with PHC.

The questionnaire had previously been tested in a pilot study based on a ten-person group of blind and poor-sighted persons. In the pilot study, the participants completed the questionnaires designed for the present survey and provided feedback on whether the questions are understandable and well-prepared for the needs of the study population.

### 2.3. Data Analyses

Statistical analyses were performed using Statistica software v. 13.3 (StatSoft, Kraków, Poland, www.stasoft.pl; accessed on 1 September 2021). Descriptive, inferential, and predictive methods were used to process the collected data. Descriptive statistics were presented for the whole PVI group and the poor-sighted and blind subgroups. Pearson’s chi^2^ test of independence was used to compare the frequency of each trait variation in the study groups. The partially sighted and blind individuals were compared with regard to their sociodemographic, health, disability, and digital-related characteristics, as well as their experience in using TCs and their willingness, motives, and needs for doing so. The associations between respondents’ characteristics and their interest in using TCs were assessed firstly with the Pearson’s chi^2^ test and then with the univariate followed by multivariate logistic analysis. Variables that were previously found to be significant, or near the significance level, were included in these analyses. Therefore, odds ratios (ORs) and 95% confidence intervals (CIs) were estimated for the following independent variables: level of VI, age, sex, education level, having a chronic disease, and internet use. Non-declared subjects, who answered “*hard to say*” when asked if they were interested in TCs, were excluded from the logistic analysis. It was not possible to conduct a logistic regression analysis among the blind respondents due to underrepresentation (*n* < 100) [47]; therefore, only the results for the total group have been provided. The significance level was at *p* = 0.05.

The responses to the open-ended questions were subjected to content analysis. All answers were recorded in a computer database. After an initial review of the replies, a framework of codes covering the subject area was created, and these were manually matched to individual statements and discussed by two coders (KBO and MAW) [48]. Some responses echoed points that appeared in answers to previous questions or comments, such as “*I have nothing to add*”, and some others were related to healthcare but beyond the scope of TM. Several replies to open-ended questions fulfilled an explanatory function for the choices explored in earlier questions. The content analysis included responses that followed the line of questioning and those that gave information about the respondents’ reluctance to use TCs. Due to the limited number of responses (a maximum of 15 assigned to one category), we performed only a descriptive analysis of answers to open-ended questions.

### 2.4. Ethics

Informed consent was obtained when completing the questionnaires. The information about the study and consent protocols were provided in paper or electronic form or read aloud before a phone interview. Respondents were informed that participation in the study was voluntary, that their consent to participate could be withdrawn at any time, and that they could ask questions to the investigator. Individuals completing the survey with the telephone interviewer or via e-mail were informed about the protection of personal data, in accordance with applicable laws. The interviewer was bounded by confidentiality. The study was conducted according to the guidelines of the Declaration of Helsinki. It was reviewed by the Bioethics Committee of the Medical University of Lodz (No. RNN/114/21IKE).

## 3. Results

### 3.1. Sample Characteristics

#### 3.1.1. Sociodemographic Characteristic of a Group

A total of 222 people participated in the survey. Answers from three respondents were excluded from the statistical analyses due no declaration of age or sex, which would make comparative analyses impossible. Nearly 60% of the participants were women. The youngest respondent was 18 and the oldest 88 years old (mean age = 50.74; SD ± 17.16). The number of participants was evenly distributed among the three age ranges: 18–39, 40–59, and 60 or older. The highest number of respondents resided in cities with more than 500.000 inhabitants, and the smallest in towns with 50–100.000 residents. The largest educational group were those with secondary education (nearly 45%), and the smallest with a basic level of education (16.7%). Almost half of those surveyed described their income as average. Only one in ten rated it as above average. One in three respondents lived with another person with a disability or need for care (Table 1).

#### 3.1.2. Health and Disability Condition of the Study Group

The study group comprised 156 poor-sighted and 63 blind participants. The prevalence of a particular type of visual disability was significantly related to gender: the male group was dominated by blind people, and the female group by poor-sighted people. More than 40% of the respondents declared having other disabilities besides VI, and this was significantly more common in the poor sighted than among the blind group: 44.9% vs. 27.0% (*p* = 0.014).

NCDs were present in almost two-thirds of the total number of respondents. They were significantly more common in the poor sighted than in the blind: 58.6% vs. 50.8% (*p* = 0.013), and in poor-sighted women compared to poor-sighted men: 73% vs. 34% (*p* = 0.039). NCDs were also more likely among older respondents than younger ones: 80.3% vs. 54% (*p* = 0.001), but less likely at higher educational levels (*p* < 0.001). Slightly more than 80% of those with multiple disabilities reported having a diagnosis of NCDs, which was almost 30% higher compared to those experiencing only VI (*p* < 0.001).

Among the group of blind respondents, only one participant reported moving independently in public spaces, whereas among those with low vision nearly 40% moved without assistance (*p* = 0.000).

#### 3.1.3. Use of TCs by Respondents in a Half Year before the Study

Nearly 70% of the study group had held TCs with a PCP within six months before the survey. The poor-sighted group was significantly more likely to have used TC than the blind group: 74.4% vs. 57.1% (*p* = 0.012), and among the partially sighted group, women had used the service more frequently than men: 80.6% vs. 63.8% (*p* = 0.020). In addition, people with NCDs consulted PCP remotely significantly more often than those without: 77.0% vs. 56.3%; *p* = 0.01 (Appendix A).

#### 3.1.4. Level of Digital Access and Skills

A significant majority of the respondents declared using computers and the internet (84.6% of the partially sighted and 90.5% of the blind). However, throughout the study group, older respondents tended to use computer equipment (*p* = 0.000) and the internet less frequently (*p* = 0.000). In contrast, the frequency of computer and internet use increased with the educational level of the participants (*p* = 0.000). Inhabitants of rural areas and cities with a population between 50 000 and 100 000 used computer equipment (*p* = 0.048) and the internet (*p* = 0.007) more often than those living in larger cities (Appendix A).

Slightly more than 75% of the poor-sighted group and 80% of the blind group rated their computer hardware skills at a medium or high level (Appendix A). About 80% of the participants in both groups rated their internet skills similarly (Appendix A). Older respondents tended to rate their computer and internet skills lower (*p* = 0.000), while those with higher levels of education tended to rate them more favorably (*p* = 0.000). In addition, participants from larger places of residence were less likely to declare high levels of internet literacy (*p* = 0.015). The level of digital competence did not appear to be influenced by gender.

### 3.2. Interest in PHC TCs among People with VI

About half of all respondents (*n* = 109) expressed an interest in using TCs in future; however, nearly 28% of the study group was not interested in making contact with a PCP in this way.

#### 3.2.1. Interest in PHC TCs and Respondents’ Characteristics

Among the blind, over 57% of the total group and nearly 60% of women and men expressed a willingness to use TCs. In the poor-sighted group, it was 45.5% of participants, with more women declaring an interest in TCs than men (50.0% vs. 37.9%), who in turn, more frequently chose the answer “*hard to say*”: nearly 40% vs. 17.35% (*p* = 0.08). The difference between poor-sighted and blind participants was not significant in this context (*p* > 0.05).

Positive attitudes toward the use of TCs decreased significantly with age: 18–39 years—57.4% willing to use it; 40–59 years—50.7%; over 60 years—38.2% (*p* = 0.014). The relationship between TCs and education levels was close to statistical significance (*p* = 0.0501), with those at a higher level of education being more willing to use TCs: 32.4% for primary education vs. 44.9% for secondary vs. 60.7% for tertiary. Additionally, among the blind, those who shared their households with other PwDs were more enthusiastic about using TCs (*p* = 0.009). No other sociodemographic variable was found to have any significant impact on interest in TCs, and no statistically significant difference was found between independent and assisted walkers with regard to the extent of interest in TCs (42% vs. 52%; *p* > 0.05).

Furthermore, having NCDs had an impact on the interest in TCs. Significantly more participants with chronic conditions reported not being interested in using TCs than those without: 32.37% vs. 20.0.% (*p* = 0.024). Having an additional disability and the use of TCs in the previous six months were also not related to the current level of interest in them (*p* > 0.05).

Similarly, no relationship was observed between interest in TCs, computer and internet use, or the self-assessment of digital competencies (*p* > 0.05). However, internet users were more likely to be interested in TCs than non-users: 51.9% and 30.0% (*p* = 0.058).

##### Interest in TCs and PVIs Characteristics in Logistic Analyses

Univariate logistic regression analysis in the whole group (Table 2), found three of the six analyzed variables to be significantly related with interest in TCs: age, level of education, and internet use. Older respondents were less likely to be interested in TCs than those from younger age groups, and those with a college education were three-and-a-half times more likely to be interested than those with a primary education. Internet users were about three times more interested in TCs than those who were not using the internet.

Multivariate logistic regression analysis (Table 2) found participants’ age to be the only factor significantly related with interest in using TCs; people aged 40–59 were almost 2.5 times more likely to be interested in using TCs than the oldest respondents (OR = 2.43; *p* = 0.048).

### 3.3. Motives for Using TCs by PVIs

The most frequently chosen motivation for using TCs was to obtain a prescription or referral (88.6%). Furthermore, half of the total group reported lacking support in getting to and from the clinic and moving around therein, as well as an increased risk of infection from other people. Nearly 37% of participants appreciated TCs during bad weather conditions. The least frequently selected reason for remote contact with PCP was the fear of loss of privacy during a conversation with a doctor in the presence of a person accompanying the patient in the clinic. Only about 10% of the people surveyed did not identify any rationale for using TCs.

Blind people were significantly more interested in remotely checking or monitoring their health than the partially sighted group: 57.1% vs. 35.9% (*p* = 0.004) and in consulting different specialists simultaneously: 44.4% vs. 21.8% (*p* = 0.001) (Figure 1).

#### Reasons for Using TCs—Respondents’ Voices

Of the respondents, 20.5% (*n* = 45) answered the open-ended question regarding the other situations they would like to use TCs with a PCP.

The content analysis of the responses revealed six additional reasons for remote contact between PVIs and PCP: (1) emergency and distressing situations, (2) need to discuss medication use, (3) difficulty in arriving at the clinic, (4) time argument, (5) obtaining sick leave, and (6) financial savings.

Regarding emergencies, respondents mentioned the need to contact a doctor in situations related i.e., to allergy, malaise, at night, and on holidays. Regarding the second motive, one participant explained why TC is beneficial to her: “*When I find that the medications I have been prescribed have a negative impact on my vision (written in the side effects), I would like to consult if there is such a risk in my case.”*

Some participants noted difficulty in arriving at the clinic associated with living far away, and the fact that TCs may be more convenient for someone assisting a PwD. In the answers assigned to the category “time argument,” respondents emphasized situations when they cannot come to an appointment due to work or study and when it is faster to travel to a TC than an inpatient visit. A few respondents wanted to request sick leave during TCs. One participant stated that TCs are cheaper because they eliminate the cost of traveling to the clinic. One respondent expressed a willingness to use TCs “*in all [situations] where it would be technically possible and equally effective.”*

### 3.4. Needs of PVIs Using TCs

The dominant need for respondents in the context of TCs with PCP was the possibility to choose between an in-person visit and TC (83.6%), followed by the physician’s communication skills (63.9%). For more than half of the respondents, having the appropriate communication equipment (smartphone, computer, and assistive equipment) was vital. The least frequently indicated need (28.8%) was the possibility for a family member/friend/assistant or another person to participate in the visit (Figure 2).

Blind participants were more likely than poor-sighted ones to propose that TCs should base on accessibility standards (54.0% vs. 37.2%; *p* = 0.022) and it should be possible for another person to be available if they are not able to notice changes in their body (46.0% vs. 29.5%; *p* = 0.019).

#### Additional Information about the Need of PVIs in the Study Group

A few supplemental areas of needs related to TCs also emerged from the content analysis of the second open-ended question (25.7%, *n* = 56) *“What else do you think is vital for the blind and partially sighted persons in the context of teleconsultation with primary care physician?”* Seven areas were selected by the coders: (1) exact date and time of TC, (2) adequate time for the visit, (3) psychosocial competencies of staff, (4) medical staff competencies related to functioning of PwDs, (5) well-designed TC, (6) awareness of the patient’s health situation, and (7) technological issues of TC. These are listed in Appendix A, accompanied by sample quotes with explanatory values (Appendix A).

Some of the responses to the open-ended question reiterated a point mentioned earlier in the replies to the closed-ended question: *“the TC should be one of the options in contacting the PCP.”* One respondent explained why it is important for her as a blind person: *“That they (TCs) don’t dominate so that a person with a disability has to leave the house. Through COVID-19 I have trouble remembering different places, my dog (guide) forgets too; I have more stress when leaving the house.”*

Although we did not ask this explicitly, several reasons for the negative attitude toward using TCs emerged from the responses to the open-ended questions. According to some respondents, TCs do not allow for sufficient conversation or examination:


*“I’m not a fan of TCs*
*—it’s impossible to examine a patient well remotely, and the interview alone isn’t everything (it’s easy for the doctor to misinterpret what the patient says).”*



*“I prefer a regular stationery visit, during which the doctor devotes more time to the patient, and over the phone sometimes the patient does not say everything he or she would like to.”*


For others, their health situation is an obstacle:


*“I think that these types of conditions (VI) require to see a doctor in person.”*



*“(...) I have hearing problems, so I don’t always understand what the doctor is saying, at an inpatient visit it is easier for me to talk.”*


Finally, also organizational difficulties discourage the use of TCs:


*“I do not like TCs—first you have to wait tens of minutes to connect to the registration and then wait on the line for tens of minutes to connect to the doctor.”*


## 4. Discussion

TM has become everyday practice in PHC during the COVID-19 pandemic and will probably be widely used in the future. Hence, there is an urgent need to explore patients’ attitudes and needs towards this form of contact with PCP. This exploration should include the population at risk of exclusion. Our study is the first such study to examine whether Polish patients with VI are interested in using TCs, to determine their reasons for doing so, and to identify their needs during remote contact with PHC.

### 4.1. Interest in PHC TCs

About 50% of the study group expressed interest in using TCs in the future. Similarly, a survey of Polish primary care patients in 2020 found that 43.2% believed that TCs should be one of the forms of contact with a PCP [39].

Among the studied characteristics, age was found to be most closely associated with willingness to use TCs. Likewise in Reed’s study [49], it decreased together with respondents’ age. This might be due to the greater reluctance to adopt innovations typical of older people and those with lower education levels [50]. In accordance with this, in the present and previous [49] studies, higher educated people and our respondents who used the internet presented more positive attitudes towards TCs. However, education level and internet utilization lost their meaning in multivariate analysis, and age was revealed as the only significant variable. It seems likely that older people were also those who used the internet less often and had lower levels of education, which weakened the effect of these variables on the willingness to use TCs, which was observed in univariate analysis. Moreover, a significantly higher interest in TCs among people at a middle age than in those at a minimum of 60 years old may be associated with the fact that the former, who are more socio-occupationally active, appreciate TCs as timesaving. People from the youngest group may have less experience in using healthcare services and less crystalized preferences than older participants, and thus this age range was not significant when variables were analyzed together. Older patients often suffer from comorbidities and visit doctors more regularly, and thus direct contact with a PCP may provide them with a feeling of safety. However, some studies have shown that patients of older age [40], including those with sensory disability [51], adapted to the virtual visits; in present study, about 14% more respondents aged 60 years or over were in favor of TCs compared with a Polish 2015 study [52]. Perhaps the development of digital literacy in recent years and the changes forced by the pandemic that have led to the widespread use of e-solutions in various areas of life could make patients more receptive to modern medical services. Hence, we believe that interventions supporting the development of technology utilization skills among older PVIs as well as empowering their belief in the effectiveness of the remote contact with PCP may change their attitudes towards TCs. Personalizing care for the elderly and those with lower education levels has been previously recommended as a solution to existing barriers; indeed, it has been demonstrated that TM improves quality of life, cognitive ability and autonomy, and reduce psychological stress among the elderly [53].

Among the sociodemographic characteristics, sex has also been previously described as significant for the interest in TM. Some studies [49,54] reported that men were less willing to use TM than women; however, this was only noted among the poor-sighted respondents in the present study. NCDs were more frequently diagnosed in the female part of the poor-sighted group, and these women might, hence, be more preoccupied with their health issues than men.

Among the blind interest in TCs was significantly associated with the presence of another person needing care in their household; this was not observed among the poor sighted. These differences between the two studied groups require further investigations.

One of the most commonly cited benefits of TM is its accessibility to people living far from medical facilities [49]. However, in the current study, the location of residence did not significantly influence the interest in using TCs. This could be due to the shorter geographic distances in Poland compared to Australia or the USA [55]. Similarly, no statistically significant difference was found in the degree of interest in TCs between independent and assisted walkers. Nevertheless, the problem of getting to a PHC (due to a long distance to the clinic or the lack of assistance) was mentioned several times in the responses to the open-ended question as an argument for using TCs. For this reason, and the fact that the access to medical services varies across Poland [55,56], further research is needed to determine the interest in TCs among PwDs living in regions with poorer healthcare, transport, and digital infrastructure.

Regarding health and disability, only having chronic conditions was found to be significantly associated with the interest in TCs: more respondents with NCDs expressed a negative attitude to TCs than those without, as mentioned previously in another Polish publication [44]. In contrast, in Bangladesh most of the respondents expressed a positive attitude towards TCs in the management of NCDs, highlighting its positive influence on mental health [57]. Although TM has been described as beneficial for people who live with NCDs [40,58,59,60], its limitations have been noted [60], and its effectiveness requires further research [61]. The nature of chronic disease is heterogeneous, and it may be difficult to monitor some of them in domestic conditions due to a lack of proper equipment. In addition, the preference for in-person visits may be influenced by patient’s habits, beliefs, and cultural context. In the light of some answers to our open questions, personal contact with PCP may be seen as more effective and enabling more comfortable communication, which is a significant consideration for those who need more regular monitoring. An interesting topic for further study would be whether the type of a disease influences the perspective of the patient towards TC use. For instance, a recent study, performed during the COVID pandemic, found TCs to be preferred by men with NCDs, but not by those with depression [62]. Also recently, Bhatia et al. [40] highlighted the need for further research to verify health conditions relevant to virtual care, and to specify the frequency, duration, and content of such visits.

Our respondents’ previous experience with TCs was not statistically significant for their current interest in using them. Previously, an international study [63] showed that TM use before the COVID-19 pandemic did not always increase the level of interest in its use after the pandemic. Further studies regarding the experiences and satisfaction with TM services among Polish PwDs could provide a deeper insight into patients’ attitudes towards TCs.

The last considered aspects in terms of willingness to use TCs were digital access and skills, as they present barriers for those attempting to use TM. Our findings indicate that only the use of the internet was significantly related with the interest in TC. It is possible that internet users may be more open to new solutions, including telehealth. Computer utilization and self-assessment of digital competencies were not significant in this regard; however, e-health solutions based on the virtual environment are not commonly used in Poland. One study published in August 2020 showed that since the beginning of the pandemic, 81.5% of participants had used TC by phone, and only 0.3% of those surveyed had taken part in a video conference with their PCP [39]. However, this could be influenced by the widening of the range of teleservices. Therefore, although in Poland access to the internet is comparable to other European countries [64] and many of our respondents declared using computer and internet, including those from rural areas, there is a need to improve digital literacy among patients by providing appropriate training and adequate facilities [65,66]. This takes on greater significance among of PwDs, since they are at risk of digital exclusion [67,68].

### 4.2. Motivations for Utilizing PHC TCs

Only 10% of participants claimed they had no reason for using TCs, while almost 30% reported having no interest in TCs. It is possible that this reluctance to use TC may be due to the lack of knowledge about its possibilities, as noted in a study of Swiss respondents [69]. PVIs have previously been found to have less awareness of the im-portance of telehealth than a control group without disability [27]. Moreover, our respondents’ recommendation to receive e-prescriptions via text message may be due to a lack of knowledge that such a possibility already exists. It may, therefore, be beneficial to confirm the degree of PwDs’ literacy regarding TM and its opportunities, and raise this level where necessary.

Our findings of the reasons for TCs utilization are consistent with the motivations described in previous studies, these being “Non-Queue Requests” such as the convenience of e-records, e-referrals, and remotely issued sick leave; time saving; emergencies [63]; and economic factors [49]. The responses to the open-ended questions indicate that TCs have been mainly regarded as useful in emergencies or, conversely, in situations where the physician knows the patient, and the consultation is of a follow-up nature. Moreover, approximately half of the surveyed group expressed a willingness to use TC when faced with, for example, SARS-CoV-2 infection. In another Polish study, almost 3/4 of the respondents indicated TC as the safest form of contact with a physician during pandemic [70]. However, since the two surveys were a year apart, it is possible that half of our respondents did not choose this option due to fatigue with remote counseling which had developed during the pandemic.

Environmental factors, such as the accessibility barriers caused by adverse weather conditions, reduced PVIs’ mobility, or the lack of support to travel to the clinic, may also influence the attitude to TCs. Adverse weather conditions make it difficult for PVIs to move around and discourage them from leaving their homes [71,72]. Although travelling without help may be troublesome, relying on help may be associated with some discomfort: PVIs generally wish to be independent and self-reliant, and this is reflected by the fact that the least frequently expressed need in our study was the opportunity to participate in TC with another person. The WHO highlights the role of environmental factors that can be either facilitators or barriers to PwDs functioning [73]. In this sense, TM allows PwDs to take care of their health without involving others and overcome the various environmental barriers to treatment mentioned before.

Finally, the blind participants were significantly more interested in monitoring their health condition via TC, and in taking part in consultations with several specialists at the same time than those with poor vision. We attribute this to their greater hardships associated with moving safely and independently.

### 4.3. Needs of Patients with VI Using TCs

While previous studies have addressed the unmet healthcare needs of patients with disabilities [74], little if any, research has related this to TM. Most of the needs identified in our study have also been confirmed in different groups of patients, as indicated below.

For almost all respondents in the present study, the core issue was that TCs should be available as an option in PHCs. This need is consistent with other studies conducted in various group of patients [75,76] and with the conclusions of a 2020 review on the use of TCs [77]. Yet, for PVIs, overuse of TCs can exacerbate their social exclusion. Many PVIs experience loneliness [78] and depression; hence, integrating them into society becomes a way to promote their mental health [79]. Indeed, one of the respondents in the present study noted the benefits of leaving the house. For this reason, personal visits to the healthcare facility should be encouraged, and the physician ought to be aware of any excessive urge to use TCs by the patient, which can be a sign of self-withdrawal and emotional disturbance.

Our respondents with low vision and blindness ex aquae emphasized the im-portance of staff communication skills and psychosocial competence including work with PwDs, as previously spotlighted concerning in-patient visits [80,81,82]. However, the current guidelines on how to communicate with the patients with VI [82] need to be revised and adapted to the realities of TM contact, and further research is needed on the impact of TC on the physician–patient relationship [83].

The third highlighted need was to have appropriate communication equipment. Technological solutions are a resource for all individuals; however, in certain groups, like PVIs, they facilitate independent functioning in various aspects of personal life, education, or work [84,85]. As mentioned before, the most common equipment used today to communicate with PHC is the smartphone. Indeed, smartphones customized for PVIs are a vital tool for inclusion in various areas of social life [86]. However, further research would be necessary to determine the influence of different media, for example using specific software to take part in video conferences, on the preferences of PVIs toward TCs.

Moreover, in the context of this study, it is important to note that peripheral devices [13,87] and Internet of Medical Technology (IoMT) [88] customized for PwDs requirements also play an important role in remote health monitoring. In all technological issues, it is still necessary to define the needs of PwDs at the design stage and develop their skills to use new technologies, as well as the skills of those who interact with them. ICT solutions should employ the principles of content accessibility contained in the so-called Web Content Accessibility Guidelines [89]. For PVIs, this entails inter alia font enlargement or text sounding. In Poland, public entities are obliged to adopt digital solutions appropriate to the diverse needs of society by the Act on Ensuring Accessibility to Persons with Special Needs [90].

While accessibility aids may enhance the comfort in using devices by some people with low vision, accessibility equals usability for blind people. Similarly, in some situations, “the use of another person’s eyes” may be their only source of information, not only about external factors but also about themselves. This may be the reason why blind respondents were more likely to specify accessibility standards and the need to receive help from someone, e.g., to notice changes in their bodies, than partially sighted participants.

Additionally, in respondents’ answers to an open question, PCP knowing their medical histories was revealed to be essential. Similarly, in another study, TC was more likely to be selected if it was to be performed by the patient’s personal PCP [49], and physicians preferred to continue care with a particular patient, claiming it is easier [91]. Furthermore, both present participants and those in the earlier studies [92,93] stated that consultations with PHC need to be longer than in-person meetings. Longer medical visits also appear to result from a patient-centered approach [92], and Cupples [82] noted that PVIs may need more time during visits for explanations and arrangements that could be forwarded in writing to non-VI patients.

An analysis of participants’ responses to the open-ended question explains their need for well-designed and punctual TCs. These are important not only for the sake of comfort, but also to allow adequate preparation for a visit and ensure proper interview conditions, including help of the PwDs assistant.

All mentioned aspects should be implemented in the curricula of medical studies to improve staff skills and the coordination of healthcare among PwDs.

### 4.4. Strengths and Limitations of the Study

The study addresses a little-explored topic with significant public health and clinical value, particularly when considering the epidemiology of VI, the proliferation of TM use in PHC, and recent changes in international and national legal health policy.

Nevertheless, our survey has some limitations. Due to the chosen sample selection method and its cross-sectional nature, its results should not be generalized, and it is impossible to determine cause-and-effect relationships. However, such non-probabilistic sampling techniques are considered reasonable for exploratory studies such as this one, whose aim is to gain an initial understanding of an issue [94], and, in this way, this paper may be a springboard to further research, including qualitative studies.

Moreover, every effort was made to ensure group heterogeneity with regard to different aspects of disability and social functioning, and to adjust the study to the specific needs connected with VI by including a pilot study, and diverse recruitment and survey methods [95]. Additionally, it allowed the project to take place during the COVID-19 pandemic, which prevented face-to-face contact with respondents.

Due to the heterogeneity of the study population and the descriptive purpose of this research, no comparison group was formed; however, the internal comparisons were made with regard to patients’ characteristics, mainly the degree of disability.

Only a portion of the group answered the open-ended questions, which also makes it impossible to generalize [96]. Nevertheless, open questions are seen as valuable in standardized surveys [97] and they enable avoiding possible bias associated with choosing predetermined answers in the closed questions [96]. In addition, analysis of responses to open-ended questions may provide practical suggestions for improving the quality of healthcare [98].

The survey took place in a specific historical context, during a pandemic that changed the way people and healthcare function. Many respondents had most likely never heard about TM before the pandemic. The next few months of the pandemic brought changes in Ministry of Health recommendations for organizing PHC services in Poland [5]. As such, attitudes toward remote services may have been influenced by respondents’ experiences and emotions, including pandemic fatigue, frustration with isolation, and fear of getting sick. In this regard, this study is worthwhile as a prelude to further investigations of the needs and attitudes of people from different groups, particularly those at risk of exclusion.

### 4.5. Implications for Practice

TM tailored to the needs of PVIs may become an essential factor in providing them with equal access to health, ensuring continuity of care and allowing them to adhere to medical appointments. Moreover, it may have positive impact on sustainable changes in healthcare policy as well as in technological, regulatory, and legislative infrastructure [13]. Hence, we recommend:involving PVIs in the process of creating health policies and auditing the accessibility of TM services to monitor if they are suitable for their needs;developing digital competencies and providing financial support to equip PVIs with user-friendly technologies for health monitoring and contact with healthcare ser-vices [99]. The elderly, internet non-users, and people with a low level of education require special attention. As the Polish population ages, the number of PwDs will inevitably increase. As such, further efforts are necessary to identify the reasons why older persons are reluctant to use TCs and to prepare people in later adulthood to use telemedical solutions;increasing health literacy in PVIs by indicating the potential of TM to make them more aware as consumers of healthcare services, expanding their care options, and thus achieving better health outcomes;improving the psychosocial and communication skills of medical staff, including awareness of biopsychosocial consequences of disability to be more open for heterogeneous needs of people. This also enables patients and increases their engagement with medical procedures [82];applying patient-centered care by respecting their preferences, needs, and values, improving their responsibility and a more active role in their health development [81,100];performing further qualitative investigations to deepen understanding of the needs of people with various disabilities in healthcare;investigating outcomes, including clinical and cost-effectiveness of TCs used in vulnerable populations like PVIs;addressing the changing needs of vulnerable populations. In response to the current humanitarian crisis in Ukraine, Poland promised free access to healthcare for Ukrainian refugees which requires the formulation of special short and long-term strategies [101,102] to include PwDs and remote healthcare services [103,104].

## 5. Conclusions

The key message from our study regarding public health, clinical practice, and patients is that TM can be a beneficial component of healthcare for PVIs if some conditions are met. It has the potential to increase their access to healthcare and improve their social inclusion. A knowledge of the biopsychosocial characteristics of PVIs and their attitudes regarding the use TCs in PHC is essential for planning health policies, healthcare coordination, and health expenditure. It provides an insight into which groups are best suited to TM, and who is best served; it also indicates what changes and conditions are necessary to engage PVIs more with TM utilization.

Some of the needs and reasons for using TCs presented by the studied PVIs may also be shared with the general population, but they are worth noting in the broader context of functioning with sensory disability. Among the study group, the blind and poor-sighted people have found several motives for using TCs in PHC, some strictly connected with the impact of disability on their functioning. However, some are still not interested in using TCs, and there is a need to find reasons for it, overcome existing barriers and increase their awareness of its benefits.

In the light of our results, special attention should be paid to older persons, those with lower education level, internet non-users, and PVIs with NCDs as they were more reluctant to use TCs. In addition, the blind and poor-sighted groups demonstrated different preferences, and these should be respected. PVIs indicated three key areas for consideration regarding the use of TCs: the optionality of TCs, the communication skills of the PCP, and appropriate equipment for communication. Failure in these areas may result in TM becoming another barrier preventing access to healthcare by PVIs. Some recommendations for implementing TM among persons with blindness and low vision, including patient-centered care, have been outlined.

In short, to answer the question posed by the title, the use of TC in PHCs appears to be beneficial for PVIs when it is optional and meets their needs. This is true not only during the pandemic, but also afterwards.

## Figures and Tables

**Figure 1 ijerph-19-06357-f001:**
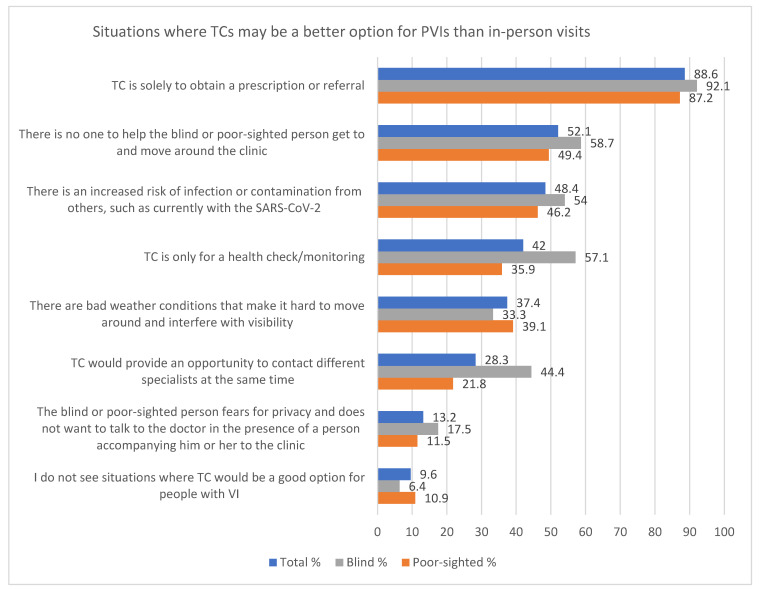
Reasons for using TCs by PVIs in percentages.

**Figure 2 ijerph-19-06357-f002:**
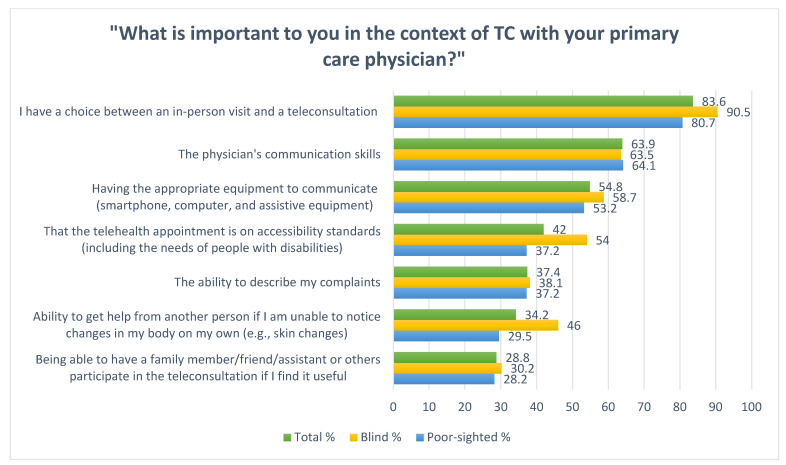
Needs of PVIs in the context of TCs with PCP in percentages.

**Table 1 ijerph-19-06357-t001:** Sociodemographic characteristic of the study group.

	Poor-Sighted *n* = 156		The Blind *n* = 63		*p*	Total	
	*n*	%	*n*	%		*n*	%
**Sex ^1^**							
Female	98	62.8	28	44.4	0.012	126	57.5
Male	58	37.2	35	55.6		93	42.5
**Age (years) ^2^**							
18–39	47	30.3	21	33.9	0.321	68	31.3
40–59	49	31.6	24	38.7		73	33.6
Over 60	59	38.1	17	27.4		76	35.0
**Place of residence ^3^**					0.913		
Rural	21	13.5	10	15.9		31	14.2
Urban to 50 k inhabitants	31	19.9	14	22.2		45	20.5
Urban from 50 to 100 k inhabitants	16	10.3	8	12.7		24	11.0
Urban from 100 to 500 k inhabitants	34	21.8	12	19.0		46	21.0
Urban from 500 k inhabitants	54	34.6	19	30.2		73	33.3
	**Education level ^4^**						
Primary educaction	26	16.7	11	17.5	0.609	37	16.9
Secondary education	73	46.8	25	39.7		98	44.7
Higher education	57	36.5	27	42.9		84	38.4
	**Level of income comparing to the average salary in Poland ^5^**						
Less than average	70	44.9	19	30.2	0.113	89	40.6
Average	71	45.5	38	60.3		109	49.8
Higher than average	15	9.6	6	9.5		21	9.6
	**Other persons with disabilities or care needs in the respondent’s household ^6^**						
Yes	43	27.6	24	38.1	0.130	67	30.6
No	113	72.4	39	61.9		152	69.4

^1^ chi^2^ = 6.202; ^2^ chi^2^ = 2.274; ^3^ chi^2^ = 0.978; ^4^ chi^2^ = 0.991; ^5^ chi^2^ = 4.367; ^6^ chi^2^ = 2.344.

**Table 2 ijerph-19-06357-t002:** Interest in TCs using in the total group—univariate and multivariate logistic regressions.

Univariate Logistic Regression				Multivariate Logistic Regression		
	OR	95%CI	*p*	OR	95%CI	*p*
**Level of visual impairment**						
The blindness	1.0	Ref.		1.0	Ref.	
The poor-sightedness	1.43	(0.70–2.88)	0.318	1.20	(0.55–2.61)	0.638
**Age (years)**						
Over 60	1.0	Ref.		1.0	Ref.	
40–59	2.72	(1.24–5.99)	**0.012**	2.43	(1.00–5.93)	**0.048**
18–39	3.07	(1.39–6.82)	**0.005**	2.35	(0.94–5.89)	0.066
**Sex**						
Male	1.0	Ref.		1.0	Ref.	
Female	1.15	(0.59–2.21)	0.683	1.01	(0.49–2.10)	0.977
**Education level**						
Primary education	1.0	Ref.		1.0	Ref.	
Secondary education	1.71	(0.69–4.24)	0.242	1.46	(0.53–3.99)	0.457
Higher education	3.50	(1.35–9.08)	**0.000**	2.68	(0.90–7.98)	0.074
**Chronic disease**						
No	1.0	Ref.		1.0	Ref.	
Yes	1.55	(0.77–3.12)	0.217	1.10	(0.50–2.43)	0.814
**Use of the internet**						
No	1.0	Ref.		1.0	Ref.	
Yes	2.95	(1.17–7.43)	**0.021**	1.19	(0.38–3.69)	0.764

*p*-values statistically significant has been bolded.

## Data Availability

The data presented in this study are available on request from the corresponding author.

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
