# Peer review of "Is Telemedicine in Primary Care a Good Option for Polish Patients with Visual Impairments Outside of a Pandemic?"

_ijerph, 2022, doi:10.3390/ijerph19116357_

Round 1
Reviewer 1 Report
Thank you very much for allowing me to review this manuscript and my sincere congratulations to the authors. The topic is very relevant, especially in the field of health inequalities. The authors indicate that this is the first study on the subject in Poland and it would be interesting to see the state of play in the rest of the world.
Methodologically I have no comments as it is well designed and executed. The limitations are also well indicated.
Here are a number of minor comments, which I believe can contribute to this study:
Abstract: it would be better to write it more concisely.
Introduction: I think the context is not sufficiently explained. During the pandemic, TM has increased in many countries, but it has also meant a misuse of TM when it has been a substitute for face-to-face care and not a complement. In fact, the possible rejection of TM can be explained by the frustration of not being able to access the health system in person. In Spain, there are social movements in favour of public healthcare which are calling for a return to face-to-face care and criticising the misuse of TM. The last sentence of your text sums it up very well (PHCs is a good option for PVIs when it is optional and meets their needs) but I would appreciate more explanation of the context in the intro.
Implications for practice: I miss some commentary on effectiveness and cost/effectiveness (especially related to the sustainability of the public health system).
Conclusions: The authors provide very interesting clues by commenting on the socio-economic reality of the users. It would be interesting to provide more aspects, such as the state of the internet in Poland (are there shadow areas without internet?).
I hope these comments are helpful and again congratulations on the manuscript.
Reviewer 2 Report
Dear author/s
Thank you for giving me the opportunity to read your manuscript. I read your manuscript with great interest. Your manuscript is well structured and organized and partially follows the instructions of the journal. Below you will find some minor points in the manuscript which need clarification, refinement, reanalysis, rewrites and/or additional information and suggestions for what could be done to improve it.
Some information and/or points are missing or unclear, and should be better included or written (e.g., objectives of the study and/or hypotheses or research questions, the type of methodology, information about the methods you used, the pilot phase, reliability, the consent protocol, etc.). To help you, here is a list of items that can be included in this section:
-What is the importance of making this research/contribution that it brings to the literature in the field?
-Why should readers be interested?
-What problem/ gap resolve/fill this research?
-To fill this gap (resolve this problem) what solution/intervention/benefits does this research bring? (In other words, how the proposed study will remedy this deficiency/gap/problem and provide a unique contribution to the literature).
-What is the research question which addresses the purpose of the research?
In summary, although some of the above are mentioned, unfortunately they are not clear. Please make the necessary adjustments. Additionally, certain sections of the manuscript begin or end abruptly, which may reduce the reader's attention or interest. I would suggest you could consider including some introductory paragraphs regarding the content of each section, in order to give the reader an idea of what to expect. Kindly check for grammatical errors, cite more of the scientific papers published by MDPI where possible and new publications that could form part of the manuscript.
Reviewer 3 Report
This study is a valuable study that evaluates the preference and utilization performance of patients with visual impairments for Telemedicine in primary care. Unfortunately, however, it is inconsistent in its objectives, analytical methods and results, and discussion and conclusions. Furthermore, the method of describing the results of the statistical analysis requires overall improvement. Please refer to the following list of points that need to be corrected. I recommend that the authors restructure this research and resubmit the paper with these points in mind.
- Objectives and Methods of Analysis
 The purpose of this study is to investigate whether PVIs are interested in using TCs in the PHC and whether this use is associated with socio-demographic variables, levels The purpose of this study is to investigate whether PVIs are interested in using TCs in the PHC and whether this use is associated with socio-demographic variables, levels of morbidity health status, previous experience in TM, and digital competencies.
 However, many of the submitted papers indicates a lot of univariate analyses of the differences between Blind and Poor-sighted; although Blind and Poor-sighted are candidates for factors that influence Interest in and TC use in PHC, the univariate analysis of the variation between Blind and Poor- Sighted univariate analysis of the variation in each factor is not consistent with the objectives of this study. In order to solve this point, the main outcomes which meet the purpose of this study should be defined in the three structured questions. For example, this study could define the main outcomes indicating interest in TC and the use of TC, and then conduct univariate and multivariate analyses to clarify the relationship between the primary outcomes and each factor. Finally, the authors should present the results in simple tables and figures.
- Discussion and Conclusion
The discussion should be based on the results of the analysis described above and should focus on comparisons with previous studies and what was revealed in this study. Although TCs is explicitly recommended in the conclusion, however, the conclusion and discussions lack consistency and rationale for the results.
- Statistical analysis
 In the description of results related to univariate analysis, instead of stating the chi2 value and p-value together, please express the p-value after indicating the parameters (%) of the two groups. In addition, the descriptions of p-values are not standardized, such as p>0.05, p<0.05, p=0,130, p<0.01, p<0.0001, P=0.0000, etc. Please unify the expressions and organize the descriptions into a Table as much as possible. Regarding the statistical analysis software, is Statistica the official name? Please indicate which company and in which country it is made.
- Strength and Limitation
This study, which surveyed opinions about TCs and their use performance from over 200 patients with visual impairment, is valuable and worth presenting. In addition, the qualitative analysis of the open-ended questions adds depth to the analysis, which we believe is a strength of this study. However, by including both quantitative and qualitative analysis, the volume of analysis is very large, and results in poor explanations of the quantitive survey. Therefore, I recommend that the qualitative analysis be included as supplementary information to support the main analysis. (i.e. Table 2 moves to Supplementary information)
Reviewer 4 Report
This paper focuses on the application and suitability of telemedicine and its services for people with visual impairments.
The paper is interesting and targets one of the recent topics which is healthcare inclusion and equity in providing services. The study has a very good sample size appropriate research methodology and design were used.
the following are recommended to enhance the quality of the paper and its readability:
1- English proofreading is highly required as in different parts, the writing and the English styles brought confusion for the readers:
- line 1
- line 50
- line 124
- line 177
2- when the authors talk about different areas, approaches or sample, it would be suggested to at least some (e.g. lines 37, 52, and 82)
3- sentence in line 50 does not seem right, as the whole point of TM is providing healthcare services outside health facilities (e.g. patients' homes) - how and why "visits to Polish 50 PHCs since the beginning of the COVID-19 pandemic were TCs"?
4- although the research design and approach are appropriate, their justification and rational of their selection is required (e.g., why cross-sectional exploratory was chosen? what methods of quantitative analysis have been done and why)
5- more information is required for the inclusion and exclusion criteria mentioned in lines 81 to 84.
6- would be great to provide a sample (as supplementary file or appendix) of The survey comprising a set of three structured questions and two open-ended questions on the subject of TM in PHC.
7- figures 1 and 2 should be moved to where the text referring to not two pages after the reference
8- these two do not seem right:
"This question allowed the respondents to develop a non-personal perspective"
and "Three questionnaires were excluded"
the whole questionnaire or just 3 questions? what was the questions' type and why had a low response rate? (mentioning the question might be helpful to know why they have been ignored by the participants)
finally, my major concern is the title and the results' analysis and discussion are not completely in line. The title of the paper implies identifying time/type/situation/or stages, having other chronic diseases or even the level and types of their VI that TM might be a good option for VI patients while the results and discussion cover the following:
- VI interest on using TM
- motivations of VI on using TM
- importance of staff communication skills and psychosocial competence
the research objective is also not in line with the topic: The study investigates whether PVIs are interested in using TCs in the PHC
Round 2
Reviewer 3 Report
I have reviewed the modifications. It has been corrected according to the review comments. I will list the points required for the final revision.
Comment1. p2 Line 49.
I guess "NCDs” is considered to stand for non-communicable diseases, not chronic diseases.
Comment2. p7 Line 311, 313.
Please correct the P value from P=0.000 to P<0.001.
Comment3. p6 Table1
As in Table 2, please list the p-values on the right side of Table 1 instead of "Total" row. It will be easier to understand which factors are statistically significant. The Chi2 score should be left outside the column as it is now.
Comment4. p13-14, line 492-519
The parts are discussed based on univariate analysis results and descriptive analyses, but do not take into account the fact that the multivariate results. Please consider the results of the multivariate analysis in your discussion. For instance, the effect of Interest in ICT almost disappeared in the multivariate analysis. This may be due to the influence of age and education level.
Reviewer 4 Report
In the second round, the authors have successfully addressed all the identified issues
Author Response
Thank you for your comments.
This manuscript is a resubmission of an earlier submission. The following is a list of the peer review reports and author responses from that submission.